# So-TVAE: Sentiment-oriented Transformer-based Variational Autoencoder Network for Live Video Commenting

## Abstract

Automatic live video commenting is with increasing attention due to its significance in narration generation, topic explanation, etc. However, the sentiment consideration of the generated comments is missing from the current methods. Sentimental factors are critical in interactive commenting, and lack of research so far. Thus, in this paper, we introduce and investigate a task, namely sentiment-guided automatic live video commenting, which aims to generate live video comments based on sentiment guidance. To address this problem, we propose a Sentiment-oriented Transformer-based Variational Autoencoder (So-TVAE) network, which consists of a sentiment-oriented diversity encoder module and a batch-attention module. Specifically, our sentiment-oriented diversity encoder elegantly combines VAE and random mask mechanism to achieve semantic diversity under sentiment guidance, which is then fused with cross-modal features to generate live video comments. Furthermore, a batch attention module is also proposed in this paper to alleviate the problem of missing sentimental samples, caused by the data imbalance, which is common in live videos as the popularity of video varies. Extensive experiments on Livebot and VideoIC datasets demonstrate that the proposed So-TVAE outperforms the state-of-the-art methods in terms of the quality and diversity of generated comments. Related codes will be released.

## 1 Introduction

Live video commenting, commonly known as "danmaku" or "bullet screen", is a new interactive mode on online video websites which allows viewers to write real-time comments to interact with others. Recently, the automatic live video commenting (ALVC) task, which aims to generate real-time video comments for viewers, is increasingly used for narration generation, topic explanation, and video science popularization as it can assist in attracting the attention and discussion of viewers. However, previous works aim to generate factual and subjective comments without considering the sentimental factor. In real-world applications, it is difficult for comments with a single sentiment to resonate with everyone (Wang & Zong, 2021; Yan et al., 2021). On the other hand, sentiment-guided comments would help video labeling and distribution, and further encourage the human-interacted comment generation. Thus, in this paper, we introduce and investigate a task, namely sentiment-guided automatic live video commenting, which aims to generate live video comments based on sentiment guidance.

Two major difficulties make this task extremely challenging. Firstly, sentiment, comment, and video are heterogeneous modalities, and they are both important in live video comment generation with sentiment. Previous works take their effort on sentiment-guided text generation (Li et al., 2021; Kim et al., 2022; Sabour et al., 2021) or video comment generation (Ma et al., 2019; Wang et al., 2020), but cannot apply to the complex situation that needs considering all three modalities simultaneously. Secondly, the imbalance of video data (Wu et al., 2021) hinders the generation of comments with the desired sentiment. The imbalance of video data lies in two aspects. On the one hand, as the popularity of the videos varies, the number of comments is a huge difference between videos, and on the other hand, in a certain video, comments usually show some sentimental trend, but lack comments on the other sentiments.

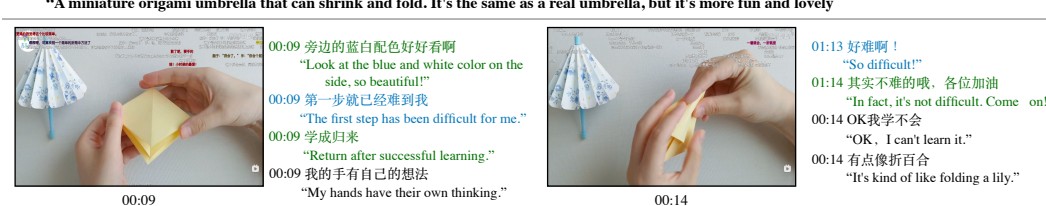

Figure 1: A live video commenting example in Livebot with selected video frames and live comments. Green: Positive comments. Black: neutral comments. Blue: Negative comments.

To this end, in this paper, we propose a Sentiment-oriented Transformer-based Variational Autoencoder (So-TVAE) network, which consists of a sentiment-oriented diversity encoder module and a batch attention module, to deal with above two challenges. The proposed sentiment-oriented diversity encoder elegantly combines VAE and random mask mechanism to achieve semantic diversity under sentiment guidance. Specifically, we firstly leverage a Gaussian mixture distribution mapping guided by sentimental information to learn diverse comments with different sentiments. In addition, to effectively avoid VAE ignoring the input information and directly learns the mapping of the latent space, we propose a novel sentiment-oriented random mask encoding mechanism, balancing the learning ability of the model to various modalities, which further improves the performance.

Moreover, a batch-attention module is proposed in this paper to alleviate the data imbalance problem existing in live video. We leverage batch-level attention along with multi-modal attention in a parallel way to simultaneously deal with the multi-modal features in batch dimension and spatial dimension. In this way, the proposed batch-level attention module can mitigate the gap between samples, explore the sample relationships in a mini-batch, and further improve the quality and diversity of generated comments.

The main contributions of this work are as follows:

Firstly, we introduce and investigate a task in this paper, namely sentiment-guided automatic live video commenting, which aims to generate live video commenting by sentiment guidance. To this end, we propose a Sentiment-oriented Transformer-based Variational Autoencoder (So-TVAE) network, which consists of a sentiment-oriented diversity encoder module and a batch attention module.

Secondly, we propose a sentiment-oriented diversity encoder module, which elegantly combines VAE and random mask mechanism to achieve semantic diversity and further align sentiment features with language and video modalities under sentiment guidance. Meanwhile, we propose a batch-attention module for sample relationship learning, to alleviate the problem of missing sentimental samples caused by the data imbalance.

Thirdly, we perform extensive experiments on two public datasets. The experimental results based on all evaluation metrics prove that our model outperforms the state-of-the-art models in terms of quality and diversity.

## 2 APPROACH

### 2.1 PROBLEM FORMULATION AND MULTI-MODAL ENCODER

Given the video clip $\mathbf{V}_t$ with $k$ frames $\mathbf{I}_t = \{I_1, \ldots, I_t, \ldots, I_k\}$ and $m$ surrounding comments $\mathbf{S}_t = \{s_1, s_2, \ldots, s_m\}$ which are firstly concatenated into a word sequence $\mathbf{x}_t = \{x_1, x_2, \ldots, x_p\}$, where $p$ represents the total number of words of $\mathbf{S}_t$, automatic live video commenting aims to generate human-like comments at timestamp $t$. The overview of the framework is illustrated in Figure 2.

Specifically, the input frames and surrounding comments are first represented as the initial features with a pre-trained convolution neural network (ResNet) He et al. (2016) and Long short-term memory network (LSTM) Hochreiter et al. (1997), respectively. Then we employ a co-attention module to further enhance feature encoding, and generate the attended visual representations $\hat{\mathbf{V}}_\mathbf{I}$ and the

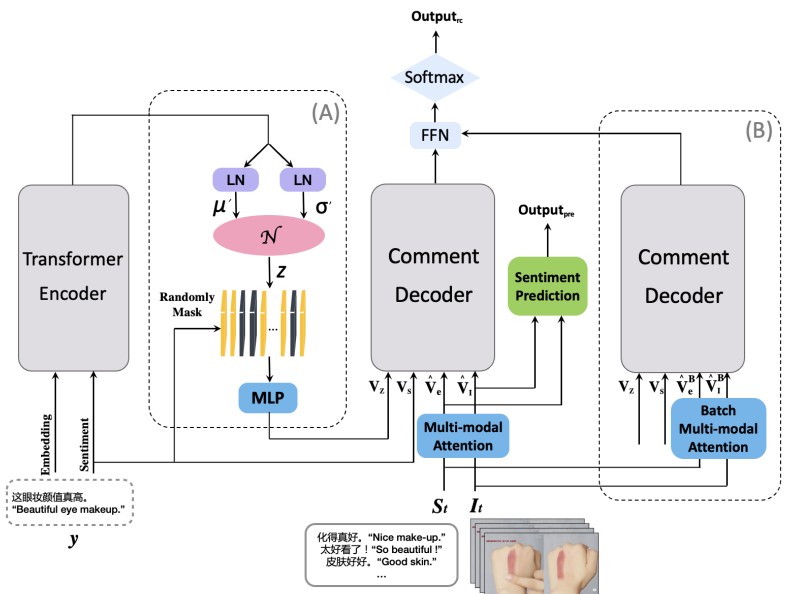

Figure 2: Overview of our proposed Sentiment-oriented Transformer-based Variational Auto-Encoder (So-TVAE) Network. (A) represents the proposed sentiment-oriented diversity encoder module; whereas, (B) represents the proposed batch attention module.

attended textual representations $\hat{\mathbf{V}}_{\mathbf{e}}$:

$$V_i = \text{ResNet}(I_i), \quad E_i = \text{LSTM}(e(x_i), E_{i-1}), \tag{1}$$

$$\hat{\mathbf{V}}_{\mathbf{e}}, \hat{\mathbf{V}}_{\mathbf{I}} = \text{Co-Attention}(\mathbf{V}_{\mathbf{e}}, \mathbf{V}_{\mathbf{I}}), \tag{2}$$

where $e(\cdot)$ is the word embedding processing. Co-attention acts on the initial frame features $\mathbf{V}_{\mathbf{I}} = \{V_1, V_2, \ldots, V_k\} \in \mathbb{R}^{B \times k \times d}$ and text feature $\mathbf{V}_{\mathbf{e}} = \{E_1, E_2, \ldots, E_p\} \in \mathbb{R}^{B \times p \times d}$ for modeling multi-modal interaction.

## 2.2 SENTIMENT-ORIENTED DIVERSITY ENCODER

The previous methods of the ALVC task can only achieve one-to-one generation as they encode the source input to a *fixed* feature vector to guide the reconstruction of comments. Therefore, based on the principle of Variational Autoencoder, we propose a diversity encoder that can explicitly model a one-to-many mapping between the latent space and the target comments. By sampling multiple latent vectors $\mathbf{z}$ from the trained latent space, our model can generate diverse comments.

### 2.2.1 SENTIMENTAL DIVERSITY

To make up for the lack of sentiment annotation in the original datasets, a pre-trained sentiment analysis model SKEP (Tian et al., 2020) is introduced to evaluate the sentiment of target comment $\mathbf{y}$, producing a sentiment label $y_s$ ($y_s \in \{0, 1, \ldots, N-1\}$, $N$ is the number of sentiment categories). We use one-hot encoding on the sentiment label to obtain sentiment weight vector $\mathbf{s}$, followed by an embedding layer to generate the direct sentiment feature representations $\mathbf{V_s} = \text{Embedding}_{\text{s}}(\text{one-hot}(y_s))$.

### 2.2.2 SEMANTIC DIVERSITY

Furthermore, we model a one-to-many mapping to learn the diverse semantic information of different sentiments. In the training stage, the semantic diversity module samples a sentiment-oriented latent vector $\mathbf{z}$ from the latent space with the guidance of target comment $\mathbf{y}$. By making $\mathbf{z}$ obey a prior probability distribution, our model can learn the diversity mapping relationship from the latent space to the reconstructed comments. Thus, in the inference stage, our model can generate diverse comments by sampling different sentiment-oriented latent vectors $\mathbf{z}$ from the prior distribution.

To effectively integrate the sentimental information and the semantic information, our model explicitly structures the latent space with the guidance of sentimental information. As the prior distribution of the latent vector $\mathbf{z}$ determines how the learned latent space is structured, which is crucial to the performance of the model, we construct it as a sentiment-based Gaussian mixture model:

$$p(\mathbf{z} \mid \mathbf{s}) = \sum_{j=1}^{N} s_j \mathcal{N}(\mathbf{z}|\mu_j, \sigma_j^2 \mathbf{I}), \tag{3}$$

where $\mathbf{I}$ is the identity matrix, $\mathbf{s}$ is the sentiment weight mentioned before, $\mu_j$ and $\sigma_j$ represent the mean vector and standard deviation of the $j$-th Gaussian distribution.

Concretely, we leverage a Transformer encoder layer to map the target comment $\mathbf{y}$ and sentiment weight $\mathbf{s}$ to a sampling region in the latent space for training. The last hidden state $h_T$ of the Transformer layer is transformed into $N$ mean vectors $\mu_j'$ and $N$ log variances $\log \sigma_j^{2'}$, using a linear network for each:

$$\mathbf{H} = \text{Transformer}(s_i, \mathbf{s}, \mathbf{y}), \tag{4}$$

$$\mu_j', \log \sigma_j^{2'} = \text{LN}_j(h_T), \quad \text{for } j = 1, ..., N \tag{5}$$

$$\mathbf{z}_{\text{post}} \sim \mathcal{N}(\mu', \sigma^{2'}; \mathbf{s}), \tag{6}$$

where $\mathbf{z}_{\text{post}}$ is the posterior latent vector, obtained by following the Gaussian mixture sampling mode of prior distribution in Equation 3.

### 2.2.3 SENTIMENT-ORIENTED RANDOM MASK

Contrary to the posterior collapse problem of VAE existing in other tasks, our model confronts a new mode imbalance problem where the model tends to rely heavily on the latent vector rather than the source input. Different from other tasks, the sentiment-guided ALVC is more difficult to reconstruct the target comments from the source input. As the latent space is infinite, the model tends to ignore the input information and directly learns the mapping between the encoding region of latent space and the reconstructed comments, which is useful in training, but meaningless at inference. Therefore, we propose a masked encoding mechanism to balance the learning ability of the model to various features by sentimental masking the posterior latent vector $\mathbf{z}_{\text{post}}$ on the random mask region $m_\lambda$. With a linear network to encode the latent vector, the multi-semantic sentiment feature representation is calculated as:

$$\mathbf{V}_{\mathbf{z}} = \text{Embedding}_{\mathbf{z}}((1 - m_\lambda) \odot \mathbf{z}_{\text{post}} + m_\lambda \odot \mathbf{z}_{\text{prior}}), \tag{7}$$

where $m_\lambda$ is a binary mask region, which regional proportion is determined by the mask ratio $\lambda$, $\odot$ is element-wise multiplication, and $\mathbf{z}_{\text{prior}}$ is the latent vector sampling from the prior distribution of the corresponding sentiment.

### 2.3 BATCH ATTENTION MODULE

To address the data imbalance problem, inspired by the batch interaction thought of Hou et al. (2022), we devise a batch attention module to explore the sample relationship during training. The proposed batch attention module leverages the batch multi-modal attention parallelly with the original multi-modal attention, which in turn leads to dealing with the multi-modal features in both batch dimension and spatial dimension. In this manner, the proposed module can mitigate the gap between samples and model their relations.

Given a batch sample $\mathbf{X} = (X_1, X_2, ..., X_B)$ with size $B$, the proposed batch attention module implicitly augments $B - 1$ virtual samples for each sample $X_i$ by modeling the relationship among samples in the mini-batch, resulting in an improvement on data scarcity. Specifically, we adopt an co-attention module acting on the input features to obtain the batch-attended textual and visual features:

$$\hat{\mathbf{V}}_{\mathbf{e}}^{\mathbf{B}}, \hat{\mathbf{V}}_{\mathbf{I}}^{\mathbf{B}} = (\text{Co-Attention}_{\mathbf{B}}(\mathbf{V}_{\mathbf{e}}^{\mathbf{B}}, \mathbf{V}_{\mathbf{I}}^{\mathbf{B}}))^{\mathsf{T}}, \tag{8}$$

where $\mathbf{V}_{\mathbf{e}}^{\mathbf{B}} \in \mathbb{R}^{p \times B \times d}$ and $\mathbf{V}_{\mathbf{I}}^{\mathbf{B}} \in \mathbb{R}^{k \times B \times d}$ are reshaped from the original frame feature and text feature, to enable the co-attention module working on the batch dimension.

In the inference stage, considering the single-sample input, the batch attention module cannot be used directly. Therefore, we introduce a new auxiliary decoder, which conducts comment decoding based on the batch-attended visual and textual features $\hat{\mathbf{V}}_{\mathbf{I}}^{\mathbf{B}}, \hat{\mathbf{V}}_{\mathbf{e}}^{\mathbf{B}}$ and the sentiment feature $\mathbf{V_s}$ and $\mathbf{V_z}$. By simply sharing weights between the auxiliary decoder and the final decoder, the model can still benefit from the sample relationship learned by the batch attention module, without utilizing the batch attention module in inference.

## 2.4 DECODER

### 2.4.1 SENTIMENT-PREDICTION

To predict the sentiment most expected by the viewers for a video clip, we design a sentiment-prediction module to predict the most likely comment sentimental types based on the visual and textual context. The predicted sentiment label $\hat{y}_s$ is calculated as:

$$\mathbf{V_{pre}} = \text{LayerNorm}(\mathbf{W_I}\hat{\mathbf{V}}_{\mathbf{I}} + \mathbf{W_e}\hat{\mathbf{V}}_{\mathbf{e}}), \tag{9}$$

$$p(\hat{y}_s \mid \mathbf{I}, \mathbf{e}) = \text{Sigmoid}(\mathbf{W_{pre}V_{pre}}), \tag{10}$$

where $\mathbf{W_I}, \mathbf{W_e} \in \mathbb{R}^{d \times d_{pre}}$ and $\mathbf{W_{pre}} \in \mathbb{R}^{d_{pre} \times N}$ are learnable weight matrixes. At inference, the sentiment label $\hat{y}_s$ predicted based on context will be used for generation, replacing the sentiment label $y_s$ from target comments.

### 2.4.2 COMMENT DECODER

The goal of the comment decoder is to generate comments for the target timestamp based on context information and diverse sentimental information. More specifically, we decode the word $\hat{\mathbf{y}}_{\mathbf{t}}$ at every timestep $t$ with a Transformer decoder layer followed by a softmax layer:

$$w_t = \text{Transformer}(y_t, \mathbf{y_{<t}}, \hat{\mathbf{V}}_{\mathbf{I}}, \hat{\mathbf{V}}_{\mathbf{e}}, \mathbf{V_s}, \mathbf{V_z}), \tag{11}$$

$$p(\hat{y}_t \mid y_0, \ldots, y_{t-1}, \mathbf{I}, \mathbf{e}, y_s, \mathbf{z}) = \text{Softmax}(\mathbf{W}w_t), \tag{12}$$

where $\mathbf{y}_{<\mathbf{t}}$ denotes the masked comment $\{y_0, \ldots, y_{t-1}\}$. Inside the comment decoder, there are five multi-head attention blocks, using $y_t$ as the query to attend to $\mathbf{y}_{<\mathbf{t}}, \hat{\mathbf{V}}_{\mathbf{I}}, \hat{\mathbf{V}}_{\mathbf{e}}, \mathbf{V_s}$ and $\mathbf{V_z}$, respectively. The auxiliary comment decoder has the same structure, but using $y_t$ as the query to attend to $\mathbf{y}_{<\mathbf{t}}, \hat{\mathbf{V}}_{\mathbf{I}}^{\mathbf{B}}, \hat{\mathbf{V}}_{\mathbf{e}}^{\mathbf{B}}, \mathbf{V_s}$ and $\mathbf{V_z}$, respectively.

## 2.5 LOSS FUNCTION

The traditional encoder-decoder model is realized by maximizing the log-likelihood function of generated comment $\hat{\mathbf{y}}$. As our generation model is controlled by latent vector $\mathbf{z}$, the objective is changed to maximize:

$$p(\hat{\mathbf{y}}) = \int_{\mathbf{z}} p(\hat{\mathbf{y}} \mid \mathbf{z})p(\mathbf{z})d\mathbf{z}. \tag{13}$$

As it is impossible to traverse all latent vectors $\mathbf{z}$ for integration, by referring to the mathematical derivation in the Variational Autoencoder (Kingma & Welling, 2014), we use the Evidence Lower Bound (ELBO) to approximate the log-likelihood function of generated comment $\hat{\mathbf{y}}$:

$$\log p(\hat{\mathbf{y}}) \geq E_{q(\mathbf{z}|\hat{\mathbf{y}})}[\log p(\hat{\mathbf{y}} \mid \mathbf{z})] - D_{KL}[q(\mathbf{z} \mid \hat{\mathbf{y}}), p(\mathbf{z})], \tag{14}$$

where $q(\mathbf{z} \mid \hat{\mathbf{y}})$ corresponds to the probability distribution obtained by the diversity encoder and is used to approximate the posterior probability distribution $p(\mathbf{z} \mid \hat{\mathbf{y}})$ of the decoder.

Since the model is also affected by the video frames $\mathbf{I}$, surrounding comments $\mathbf{e}$, and generated or selected sentiment type $y_s$, the objective function is changed to:

$$\log p(\hat{\mathbf{y}} \mid \mathbf{I}, \mathbf{e}, y_s) \geq E_{q(\mathbf{z}|\hat{\mathbf{y}}, \mathbf{I}, \mathbf{e}, y_s)}[\log p(\hat{\mathbf{y}} \mid \mathbf{z}, \mathbf{I}, \mathbf{e}, y_s)] - $$
$$D_{KL}[q(\mathbf{z} \mid \hat{\mathbf{y}}, \mathbf{I}, \mathbf{e}, y_s), p(\mathbf{z} \mid \mathbf{I}, \mathbf{e}, y_s)]. \tag{15}$$

The first item encourages the model to generate higher quality comments, which is the loss of model reconstruction. The second item encourages the latent vector distribution obtained by training to be

as close as possible to the prior distribution $p(\mathbf{z} \mid \mathbf{I}, \mathbf{e}, y_s)$, i.e., $p(\mathbf{z} \mid y_s)$, which is the KL distance between them:

$$loss_z = D_{KL}[q(\mathbf{z} \mid \hat{\mathbf{y}}, \mathbf{I}, \mathbf{e}, y_s), p(\mathbf{z} \mid y_s)]$$

$$= \sum_{j=1}^{N} \log(\frac{\sigma_j}{\sigma_j'}) + \frac{\sigma_j'^2 + \| \mu_j' - \mu_j \|_2^2}{2\sigma_j^2} - \frac{1}{2}. \qquad (16)$$

Moreover, considering the sentiment-prediction module, we adopt cross-entropy loss to optimize:

$$loss_{pre} = -\log p_\theta(\hat{y}_s \mid y_s), \qquad (17)$$

where $\theta$ corresponds to model parameters, $\hat{y}_s$ and $y_s$ are the predicted sentiment and ground truth sentiment respectively.

In short, the total loss function of the model is defined as:

$$\mathcal{L} = loss_{rc} + \beta \cdot loss_z + \gamma \cdot loss_{pre}, \qquad (18)$$

where $\beta$ and $\gamma$ are hyper-parameters.

## 3 EXPERIMENT

**Experiment Setting.** We conduct experiments on two live video commenting datasets: Livebot (Ma et al., 2019) and VideoIC (Wang et al., 2020). In all experiments, we consider the Transformer layers with 6 blocks, 8 attention heads, and set the hidden size of the multi-head attention layers and the feed-forward layers to $512$ and $2,048$. Our model is optimized by Adam optimizer (Kingma & Ba, 2014) with the base learning rate set to $1e-4$ and $3e-5$ on datasets Livebot and VideoIC respectively, and decayed by $1/4$ every 2 epochs after 4 epochs. The batch size is $128$ and dropout rate is $0.1$. The number of selected video frames and surrounding comments $k$ and $m$ are both set to 5. For the joint loss function, the corresponding parameters are set as: $\beta = 2$ and $\gamma = 0.3$. We adopt 3 as the number of layers of the co-attention module. In practice, the standard deviation of each prior distribution is fixed at $0.2$. Unless otherwise stated, the sentiment category $N$ is set to 3, as $N = 3$ exactly corresponds to the three sentiments of negative, neutral, and positive.

**Evaluation Metrics.** Since a time stamp contains multiple various comments, which makes the reference-based metrics not applicable, we use the retrieval-based evaluation metrics proposed by Ma et al. (2019) for the quality evaluation. As the evaluation is formulated as a ranking problem, the model is asked to rank a set of candidate comments based on the log-likelihood score. We provide a candidate comment set that consists of diverse ground-truth comments and three types of improper comments as follows: 1) **Correct**: the ground-truth comments of the corresponding video and corresponding moment provided by human. 2) **Plausible**: 50 comments most similar to the video title which are measured based on the cosine similarity of their TF-IDF values. 3) **Popular**: 20 most popular comments from the dataset which carry less information, such as "Great", "hahaha". 4) **Random**: after selecting the correct, plausible, and popular comments, randomly select some comments to make sure the number of the candidate set contains 100 comments.

The model ranks the ground-truth comments higher with a higher generation probability, reflecting better model performance. We measure the ranking results with the following metrics: **Recall@k** (the proportion of ground-truth comments in the top-k sorted comments), **Mean Rank** (the mean rank of the ground-truth comments), **Mean Reciprocal Rank** (the mean reciprocal rank of the ground-truth comments).

### 3.1 COMPARISON WITH STATE-OF-THE-ART

In order to verify the effectiveness of the proposed model, we conduct experiments to compare our So-TVAE with the state-of-the-art methods, including Fusional RNN (Ma et al., 2019), Unified Transformer (Ma et al., 2019), MML-CG (Wang et al., 2020), DCA (Zhang et al., 2020b) and Matching Transformer (Duan et al., 2020). Besides, to further explore the impact of sentiment, we conduct the experiments with different sentiment categories (3, 5, and 10).

Table 1 shows the comparison results. Our proposed So-TVAE exhibits better performances on both two datasets. It is worth noting that our sentiment-oriented model does not simply improve as the

Table 1: Performance comparisons with the state-of-the-art methods, where R@k, MR, MRR, are short for Recall@k, Mean Rank and Mean Reciprocal Rank. $\downarrow$ indicates that the performance is better with a lower score. $_{3,5,10}$ annotates the results of setting $N = 3, 5, 10$.

| Model | Livebot | | | | | VideoIC | | | | |
|---|---|---|---|---|---|---|---|---|---|---|
| | R@1 | R@5 | R@10 | MR$^{\downarrow}$ | MRR | R@1 | R@5 | R@10 | MR$^{\downarrow}$ | MRR |
| F-RNN | 17.25 | 3796 | 56.10 | 16.14 | 0.271 | 25.38 | 51.02 | 64.36 | 11.73 | 0.400 |
| U-Trans | 18.01 | 38.12 | 55.78 | 16.01 | 0.275 | 26.34 | 54.66 | 64.37 | 12.66 | 0.390 |
| MML-CG | 18.57 | 43.80 | 54.09 | 15.37 | 0.312 | 27.50 | **56.12** | 65.68 | 12.21 | 0.402 |
| DCA | 25.80 | 44.20 | 58.40 | 15.10 | 0.353 | – | – | – | – | – |
| M-Trans | 22.71 | 46.71 | 62.87 | 11.19 | 0.352 | – | – | – | – | – |
| So-TVAE$_3$ | 25.88 | 50.64 | 65.68 | 11.10 | 0.384 | 29.58 | 55.35 | 69.67 | 9.538 | 0.413 |
| So-TVAE$_5$ | **27.07** | **52.97** | **67.44** | **10.54** | **0.397** | 27.50 | 54.73 | 69.93 | 9.632 | 0.407 |
| So-TVAE$_{10}$ | 24.04 | 51.46 | 66.83 | 10.73 | 0.374 | **29.59** | 55.09 | **70.34** | **9.431** | **0.415** |

Table 2: Human evaluation result on the Livebot test split.

| Model | Flu. | Rel. | Cor. | Div. |
|---|---|---|---|---|
| F-RNN | 2.44 | 2.52 | 2.40 | 1.32 |
| U-Trans | 3.44 | 3.28 | 3.52 | 1.48 |
| MML-CG | 3.82 | 3.44 | 3.84 | 1.42 |
| So-TVAE | **4.36** | **3.72** | **4.32** | **3.88** |
| Human | 4.64 | 4.12 | 4.52 | 4.24 |

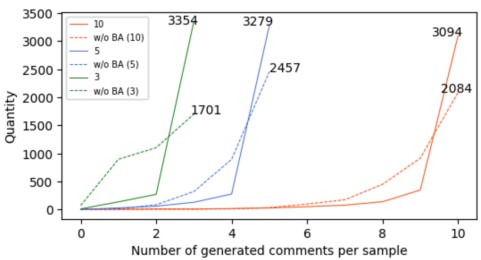

Figure 3: Quantity distribution curve of meaningful generated comments.

increase of sentiment categories. The model achieves the best performances on Livebot dataset with $N = 5$ and VideoIC dataset with $N = 10$, meaning that the best effects will be obtained when the fine granularity of sentiment is matching the sentimental distribution of the dataset.

**Human Evaluation.** We test the model ability of generating human-like comments. The generated comments are evaluated from four aspects: **Fluency** (fluency of comments), **Relevance** (relevant comments to video content), **Correctness** (the confidence that the comments are made by humans), and **Diversity** (diversity of multiple generated comments). We randomly select 200 video clips, which include $1,000$ comments from three baseline models, our So-TVAE, and ground-truth. Three human annotators are asked to score the selected instances on four metrics in the range of $[1, 5]$, with the higher the better. Finally, we take the average score of all annotators as the human evaluation results. As can be seen from Table 2, our proposed So-TVAE achieves better performance in all aspects, especially in fluency, correctness and diversity. The great improvement in correctness indicates that our model can generate more human-style comments.

**Qualitative Results.** Figure 4 exhibits several live video commenting results of the baseline models and our So-TVAE, coupled with the video frames and contextual comments. Generally, compared with the baseline model, our So-TVAE produces more accurate and detailed comments. Taking the image of Figure 4 (b) as an example, our model generates a more detailed and accurate phrase "tired", while the baseline models only predict a high-frequency phrase "the first day". On the one hand, as we can see from Figure 4, by choosing different sentimental types, the current model can successfully generate comments of different sentiments for a single video frame, which proves the effectiveness of the sentimental diversity module. On the other hand, after selecting the same sentimental type, our model can generate comments of different semantics with the same sentiment, such as "Really good" and "The harmony is so great" in part (d) of Figure 4, verifying the effectiveness of the semantic diversity module. In general, the sentiment-oriented multi-semantic comment generation method is more in line with human thinking habits, resulting in more humanized comments with the language characteristics of multiple perspectives and multiple sentiments.

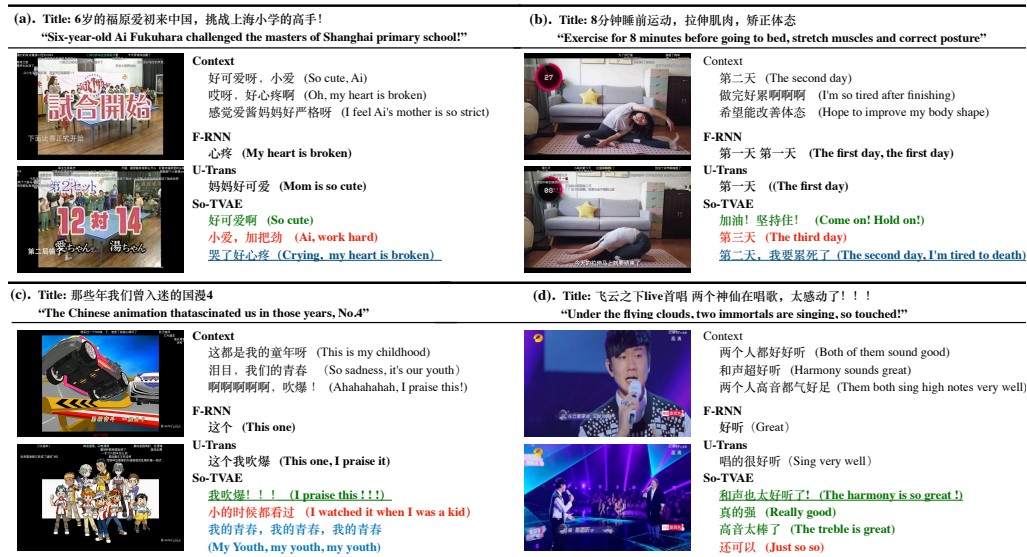

Figure 4: Examples of generated comments by the baseline and So-TVAE for live video commenting. Green: positive comments. Red: neutral comments. Blue: negative comments. Context: human posted comments in context. The underline marks the generated comments of predicted sentiment.

Table 3: The results of ablation studies on Livebot and VideoIC.

| Module | Variant | Livebot | | | | | VideoIC | | | | |
|---|---|---|---|---|---|---|---|---|---|---|---|
| | | R@1 | R@5 | R@10 | MR$^\downarrow$ | MRR | R@1 | R@5 | R@10 | MR$^\downarrow$ | MRR |
| Diversity | w/o | 24.34 | 47.21 | 62.47 | 12.36 | 0.363 | 28.44 | 52.10 | 66.39 | 10.73 | 0.399 |
| | SEND | 24.39 | 47.24 | 62.83 | 12.61 | 0.362 | 28.73 | 53.48 | 67.70 | 10.30 | 0.412 |
| | SMD | 25.22 | 50.50 | 64.76 | 11.65 | 0.382 | 29.20 | 54.47 | 68.41 | 10.10 | 0.418 |
| Mask | $\lambda=0\%$ | 24.50 | 49.18 | 63.96 | 11.97 | 0.370 | 28.48 | 54.44 | 67.87 | 9.964 | 0.403 |
| | $\lambda=15\%$ | 25.21 | 50.42 | 64.76 | 11.48 | 0.379 | 29.22 | 55.27 | 68.89 | 9.798 | 0.421 |
| | $\lambda=45\%$ | 24.84 | 50.61 | 65.60 | 11.28 | 0.378 | 29.05 | 55.08 | 69.40 | 9.614 | 0.419 |
| Batch-Attention | w/o | 21.39 | 45.12 | 59.66 | 12.83 | 0.336 | 21.77 | 45.57 | 59.98 | 12.76 | 0.342 |
| | Origin | 22.93 | 47.64 | 61.94 | 12.33 | 0.354 | 29.00 | 53.42 | 67.43 | 10.55 | 0.412 |
| Co-attention | w/o | 8.678 | 20.04 | 34.34 | 22.12 | 0.171 | 16.98 | 38.08 | 53.26 | 15.69 | 0.284 |
| Full model | So-TVAE | **25.88** | **50.64** | **65.68** | **11.10** | **0.384** | **29.58** | **55.35** | **69.67** | **9.538** | **0.413** |

## 3.2 ABLATION STUDY

To fully examine the contribution of each design in our model, we conduct ablation studies to compare different variants of So-TVAE. The corresponding variants of each module are shown below:

**Diversity Encoding**: 1) the base model without any diversity components; 2) SMD: uses the standard normal distribution as the prior distribution without sentimental information; 3) SEND: simply encodes the sentimental information as the input to the decoder.

**Mask**: the models with different mask radio $\lambda$, where $\lambda = 0$ means without masked encoding.

**Batch Attention** [1]: 1) the base model without any batch attention module; 2) the model which uses the original BatchFormer module proposed in Hou et al. (2022) to model the sample relationship.

**Co-attention** [2]: the model without co-attention module.

According to the results shown in Table 3, we have the following observations: (1) For diversity encoding, both the semantic diversity module and the sentimental diversity module bring an improvement with respect to the base model. (2) When the mask ratio $\lambda$ is too large or too small, the

---

[1]See Appendix B for more details.

[2]See Appendix A for more details.

model will not learn enough source input information or diversity information. By setting the appropriate mask ratio to capture a balance between diversity and average quality, the model achieves the best overall performance with $\lambda = 30\%$ (full model). (3) The model with proposed batch attention module substantially outperforms the model without batch attention or with the original module, proving the effectiveness of our reconstruction of sample relational learning.

In addition, to further demonstrate the effectiveness of the proposed batch-attention module on data imbalance problems, we carry out extensive experiments. For the sentiment-guided live video commenting, an unfavourable situation of data imbalance is that with the absence of comment samples with corresponding sentiment, the model is difficult to learn this type of comment information and tends to generate an empty sentence. Thus, we use the model with and without batch attention module to generate $N$ (3, 5, and 10) comments for all 3,768 samples in the Livebot test split, exploring the distribution of meaningful comment numbers. Figure 3 shows the distribution results. Obviously, we can observe that the model with batch-attention module can generate more meaningful comments than without utilizing batch-attention module. Taking the 10 sentiments situation as an example, 3,094 samples can generate comments for each sentiment using the full model, while only 2,084 samples can generate all sentimental comments without batch-attention. These results further prove that our proposed batch-attention module can effectively alleviate the problem of data imbalance in sentiment-guided ALVC.

## 4 RELATED WORK

Similar to but different from video captioning (Pan et al., 2020; Tan et al., 2020), image captioning (Zhang et al., 2020a; Zhao et al., 2020) and diverse text generation (Dathathri et al., 2020; Nie et al., 2019), automatic live video commenting aims at generating interactive comments, requiring a fully understanding of surrounding multimodal context. Ma et al. (2019) first proposed the ALVC task and proposed two basic neural networks to jointly encode the visual and textual information into video comments. Duan et al. (2020); Zhang et al. (2020b) explored the interaction between multimodal information. Wang et al. (2020) constructed the VideoIC dataset on the consideration of higher comment density and diversity, and proposed a multimodal and multitask generation model to achieve effective commenting and temporal relation prediction. Zeng et al. (2021); Wu et al. (2021) further explored highlight detection and temporal interaction for ALVC. However, these works ignore the importance of multi-sentiment in the application of live video commenting.

To achieve the sentiment-guided live video commenting, we consider some one-to-many generation methods for paired data, such as the mixture of experts (Shen et al., 2019), discrete domain encoding (Lachaux et al., 2020; Gan et al., 2017) or Variational Autoencoder (Kingma & Welling, 2014; Child, 2021; Zhu et al., 2020; Shao et al., 2020). Among these, Variational Autoencoders were proposed by Kingma & Welling (2014); Rezende et al. (2014), and were initially applied in the computer vision task. Bowman et al. (2015) introduced VAEs into natural language processing for text generation for the first time, then VAEs have been extended by many following works in various language generation tasks (Wen et al., 2017; Zhao et al., 2018; Hu et al., 2017). To date, Conditional Variable Autoencoders (CVAEs) have been used in image generation (Child, 2021), text summarization Wang et al. (2019), image question generation Jain et al. (2017), image caption (Wang et al., 2017), dialog model (Zhao et al., 2017) and other tasks, our work is the first to apply them to the ALVC task. By addressing the problem of multi-modal learning and sentiment information fusion, our model achieves the generation of diverse comments under sentiment guidance.

## 5 CONCLUSION

In this paper, we propose a Sentiment-oriented Transformer-based Variational Auto-Encoder (So-TVAE) for the sentiment-guided ALVC task which can achieve sentiment-guided live video commenting. Specifically, we propose a sentiment-oriented diversity encoder to model the rich semantic diversity information and the sentimental diversity information in the target comments. To alleviate the mode imbalance problem, we propose a novel sentiment-oriented random mask mechanism to ensure the quality and diversity of the generation model. In addition, we also propose a batch attention module to model the sample relationship, which is useful to address the video data imbalance problem.Extensive experiments verify the effectiveness of our proposed method.

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
