# OpenReview forum: "So-TVAE: Sentiment-oriented Transformer-based Variational Autoencoder Network for Live Video Commenting"
_ICLR.cc/2023/Conference — Submitted to ICLR 2023_

### Official Review · Reviewer_5558 · 2022-10-23

**Confidence:** 3
**Clarity, Quality, Novelty And Reproducibility:** N/A
**Correctness:** 4
**Technical Novelty And Significance:** 2
**Empirical Novelty And Significance:** 2
**Recommendation:** 5

**Strength And Weaknesses:**

Strengths:
1. Novel and interesting task.
2. Well designed model, and achieve state-of-the-art performance on relative datasets.

Weaknesses:
1. It’s a bit unclear when introducing the motivation. According to Section 2.1, this task aims to generate the new automatic comments based on the context of some historical comments. Naturally, the historical comments may have multiple sentiments. Then, what kind sentiments should be used to guide generation of new comments, and why?

2. Why does Equation 4 has three inputs. According to Figure 2, the Transformer Encoder takes $s$ and $y$ as inputs.

3. In Equation 18, the denote $L_rc$ should be pre-defined.


**Summary Of The Paper:**

The paper proposes a Sentiment-oriented Transformer-based Variational Autoencoder (SO-TVAE) model for the automatic live video commenting task. The SO-TVAE takes sentiment clues of the surrounding comments to generate new automatic live video comments. The model also adopts the VAE to achieve commenting sentences with diverse sentiments and semantics.

**Summary Of The Review:**

The paper studies a novel and interesting task ALVC. A new model is proposed to solve the sentiment and semantic diversity problem in ALVC, which achieves significant improvement compared with previous works.

---

### Official Review · Reviewer_b3zH · 2022-10-24

**Confidence:** 5
**Correctness:** 2
**Technical Novelty And Significance:** 2
**Empirical Novelty And Significance:** 2
**Recommendation:** 3

**Clarity, Quality, Novelty And Reproducibility:**

The paper itself is easy to read and its clarity is medium,
but the approach, its description and evaluation is vague and insufficient,
so its quality and novelty cannot be assessed at this stage.
Reproducibility is not high for those familiar with this field, as long as the paper's description is accurate.

**Strength And Weaknesses:**

Strength\
*Automatic live video commenting is interesting task.

Weaknesses\
*Low technical novelty of model\
The model is a straightforward collection of known frameworks (Transformers and VAE),
its motivation is unclear, and the framework itself is a bit dated.
VAE and random mask mechanism may solve the semantic diversity under sentiment guidance,
but how does this mechanism or batch-level attention along with multi-modal attention solve the modality gap between text and video?\
Is there any reason to use Transformer and LSTM, as the initial features layer, together instead of just Transformer?
etc.,

*Technical terms and descriptions are incomplete\
If $s_{i}$ is the weight of $I$-th sentiment?, what makes $N()$ in eq(3) different than $N()$ in eq (6) ?\
Where is  $W_{w_{t}}$  and $L_{rc}$ defined?
Is there any reason why the random weight is more efficient than learnable weights in eq (7)?

*Lack of existing research on Transformer-based models  and the resulting distrust of ideas for proposed architecture.\
Why is there no research and comparison on the application of pre-trained models like GPT-2 as the decoder?\
This lack leads to the weakness of the comparative experiments,
and can not support the theoretical basis of the approach.

*The purpose and results of the experiment do not adequately support the claim.\
Table 2 show that an assessment of the consistency of the comments as man-made comments, but does not appear to be an assessment of whether they reflect emotion.\
What are the video categories and their ratios?\
How much is the ratio of positive-negatives in the data set?\
As the proposed model is a generative model but does not use METEOR, ROUGE, or BLEU as the valuation metrics.

**Summary Of The Paper:**

This paper a Sentiment- oriented Transformer-based Variational Autoencoder (So-TVAE) , which consists of a sentiment-oriented diversity encoder module and a batch-attention module.
The encoder part combines VAE and random mask mechanism to achieve semantic diversity under sentiment guidance, which is then fused with cross-modal features to generate live video comments,
while the batch-attention module is to alleviate the problem of missing sentimental samples.
Authors perform experiments on Livebot and VideoIC datasets and compare it with the state-of-the-art methods in terms of the quality and diversity of generated comments.

**Summary Of The Review:**

The authors propose and evaluate a Sentiment-oriented Transformer-based Variational Autoencoder (So-TVAE) network in this paper to achieve sentiment-guided live video commenting.
However, the novelty and theoretical basis of the proposed model and its evaluation are insufficient.

---

### Official Review · Reviewer_TAXK · 2022-10-25

**Confidence:** 4
**Clarity, Quality, Novelty And Reproducibility:** 1. The paper clarity still needs impr…
**Correctness:** 3
**Technical Novelty And Significance:** 2
**Empirical Novelty And Significance:** 2
**Recommendation:** 5

**Strength And Weaknesses:**

Strength:
The authors work on an important problem and the proposed method is able to introduce sentiment information into complicated live video comment generation.  Additionally, the authors propose a new batch attention mechanism to further improve the performance.

Weakness:
1. The authors hypothesize the important of the sentiment information. However, such a motivation is not clearly supported by the empirical studies. In the table 3,  SMD (uses the standard normal distribution as the prior distribution without sentimental information) is able to provide a very close performance to the proposed method. Such a result cannot support the motivation of the proposed method, which is sentiment-guided automatic live video commenting.
2. The improvement gap brought by the proposed method are not very large. The multiple runs are needed to validate the significance.
3. The technical novelty is limited. The proposed method is build upon combination of the existing techniques without clear novel designs.
4. In the ablation studies, some of the results from the ablation baselines are better than the proposed method, like the last column in VideoIC, several values  in term of MRR are higher than the bolded value 0.413.


**Summary Of The Paper:**

The authors identify the importance of the sentiment information in automatic live video comments. Toward this end, the authors work on sentiment-guided automatic live video commenting, which aims to generate live video comments based on sentiment guidance.  More specifically, they propose a Sentiment-oriented Transformer-based Variational Autoencoder (So-TVAE) network, which consists of a sentiment-oriented diversity encoder module and a batch-attention module.

**Summary Of The Review:**

The authors work on sentiment guided live video commenting. The problem is interesting and challenging. However, the motivation regarding sentiment is not well supported by the empirical results. More details can be referred to Strength And Weaknesses section.

---

### Official Review · Reviewer_bz3b · 2022-11-02

**Confidence:** 4
**Correctness:** 3
**Technical Novelty And Significance:** 3
**Empirical Novelty And Significance:** 3
**Recommendation:** 6

**Clarity, Quality, Novelty And Reproducibility:**

The overall pipeline has novelty in the way encoder and the decoders are connected. There are some blocks not shown in the diagram (for example the object detectors and the LSTMs) which makes it a bit hard to understand the data flow in the overall pipeline.

**Strength And Weaknesses:**

Weaknesses:
- The paper introduces this task as a feature of interactive mode online video websites. The authors mention that that this feature is increasingly being used on websites for narration generation etc. Some references to the use cases will be helpful to underscore the relevance of this task.
- Authors introduce 3 contributions in the paper: the contributions on introducing a new task as well as extensive experiments are not really novel contributions (given that the task is ALVC and running experiments and evaluation isn't really a novel contribution of the work)
- Authors do not discuss the inference times for running the various VAE based models as well as the sentiment prediction models. Can this method be used efficiently at inference time?

Strengths:
- There isn't novelty in the individual algorithms proposed but the overall pipeline that includes blocks on sentiment diversity encoder, the batch level auxiliary decoder and the combination of the encoder and decoder using VAEs looks very elegant.
- Experiments are conducted on 2 datasets with extensive evaluation (automatic and human) and top of line results.


**Summary Of The Paper:**

This paper presents methods for the task of Automatic Live Video commenting (ALVC). The paper proposed a sentiment oriented robust diversity encoder and a batch level decoder network that can handle missing sentiment samples in the dataset. The authors conduct extensive experiments on two datasets and demonstrate that this method outperforms other known methods for automatic live video commenting.

**Summary Of The Review:**

Overall, the paper is pretty well-written for the task of ALVC. The proposed sentiment-oriented diversity encoder combines VAE and random mask mechanism to achieve semantic diversity under sentiment guidance. The paper also uses a batch-attention module to alleviate the data imbalance problem. The work leverages batch-level attention along with multi-modal attention in a parallel way (using auxiliary tied together decoder model) to simultaneously deal with the multi-modal features in batch dimension and spatial dimension. Results and ablations show that the method is promising for the problem.

---

### Decision · Program_Chairs · 2023-01-20

**Decision:**

Reject

**Justification For Why Not Higher Score:**

The submission is not of ICLR quality.

**Justification For Why Not Lower Score:**

As described above.

**Metareview: Summary, Strengths And Weaknesses:**

Strength
* A new model is proposed for the task.
* Experiments are conducted on two datasets.

Weakness
* A new task is proposed. However, justification of the proposal is not sufficient.
* The novelty is limited. The proposed method is a combination of existing techniques.
* The experimental results need to be more convincing. The improvements in the method need to be more significant. Some of the ablation baselines are better than the proposed method.
* The empirical results do not fully support the motivation.
* The presentation can be improved. There are technical details that are not clearly described.

**Summary Of Ac-Reviewer Meeting:**

There was no response from the authors.

I initiated a discussion. There was no reply from the reviewers.